# Prediction of Tool Wear Using Artificial Neural Networks during Turning of Hardened Steel

**DOI:** 10.3390/ma12193091

**Published:** 2019-09-22

**Authors:** Paweł Twardowski, Martyna Wiciak-Pikuła

**Affiliations:** Faculty of Mechanical Engineering and Management, Poznan University of Technology, 3 Piotrowo St., 60-965 Poznan, Poland

**Keywords:** artificial neural network, prediction, tool wear

## Abstract

The ability to effectively predict tool wear during machining is an extremely important part of diagnostics that results in changing the tool at the relevant time. Effective assessment of the rate of tool wear increases the efficiency of the process and makes it possible to replace the tool before catastrophic wear occurs. In this context, the value of the effectiveness of predicting tool wear during turning of hardened steel using artificial neural networks, multilayer perceptron (MLP), was checked. Cutting forces and acceleration of mechanical vibrations were used to monitor the tool wear process. As a result of the analysis using artificial neural networks, the suitability of individual physical phenomena to the monitoring process was assessed.

## 1. Introduction

Currently, many methods are used to evaluate tool wear in real time. The cutting process monitoring system is a tool used to eliminate catastrophic tool failure (CTF). Assessment of the condition of the tool wear based on physical quantities that are associated with the cutting process is possible on the basis of many different methods. Research has been conducted over the years, which compares the effectiveness of assessing tool wear condition based on diagnostic inference methods. Regression models and pattern recognition are among the basic methods. Monitoring of the manufacturing processes is an issue that still requires improvement, despite the use of many modern systems in industry. One of the newer methods used to monitor the condition of the cutting edge is the empirical method, empirical mode decomposition (EMD), which is based on the decomposition of signals in the time domain. As reported in Olufayo et al. [1] paper, it was used to detect the cracking of the tool based on the measurement of cutting forces. The researchers presented an online industrial monitoring system that reliably obtained precise information on tool wear. On the basis of the coefficient of friction and average power, two forms of wear, i.e., tool edge chipping and tool edge wear, were detected in real time using the CUSUM algorithm (cumulative sum control chart), which gave satisfactory results as compared to an offline method. Wide usage in machining also has indirect monitoring of cutting wear based on cutting forces, evaluation of chip morphology, mechanical vibrations, and acoustic emission. An internet system for measuring and monitoring tool wear based on machine vision was designed and developed in line with the characteristics of a ball-end cutter. The validation of the experiment showed an error of only 2.5% in relation to the actual tool wear. In Wang et al. [2] investigations, acoustic emission signals were used to diagnose ceramic inserts during milling at high cutting speeds, where a multisensor system for classifying the used cutting wedge was additionally employed. The research used spectral analysis observation and wavelet feature extraction to evaluate tool wear. On the basis of the data obtained, the researchers developed a feed-forward backpropagation neural network (BPNN) model to predict tool wear. Learning the neural network gave a low error value of 0.00523 to classify the state of the tool. This is a very low average square error relative to actual consumption values, which confirms the effectiveness of the prediction based on acoustic emissions. Very often, cutting forces are used to diagnose the condition of the tool, and neural networks are used for diagnostic inference [3,4]. In addition to neural networks, the wavelet transformation and spectral grouping algorithms are also used for diagnostic inference such as in Aghazadeh et al. [5] paper. This experiment presented a tool conditioning monitoring (TCM) system using deep convolutional neural networks (CNNs) as an effective method of deep learning. Force and vibration signals from the experimental ETS (Emissions Trading System) dataset were used, which were independently selected to develop a monitoring system. In contrast to other learning models, these data-driven models were able to learn discriminative nonlinear feature representations. In this way, they could provide an efficient prediction model for error detection by learning feature representations directly from the input signals. Different methods of machine learning algorithms with force and vibration signals were compared and the smallest root mean square error (RMSE) was obtained for the CNN (0.0709 for force signals and 0.086 for vibration). Another example is the research of Kong et al. [6] paper that presented an effective model of wear width prediction using the Gaussian model with the radial basis function kernel principal component analysis (KPCA_IRBF). In this technique, the Gaussian noises can be modeled quantitatively in the GPR (Gaussian Process Regression) model. Many studies confirm that cutting forces are the most sensitive to changes in tool wear. However, their industrial application involves interference in the construction of machine tools or causes restrictions in the working space. Vibration sensors do not have such limitations, and therefore they are easy to assemble and do not interfere with the machine’s construction. Therefore, diagnostic methods based on vibration measurements are constantly developed. One of the solutions is the use of multisensors because different sensors correlate better with subsequent stages of tool wear. This solution gives a full view of potential wear. After receiving the raw signal, signal processing and feature extraction methods are used, i.e., time domain analysis using autoregressive models (AR), moving average models (MA) or autoregressive moving average (ARMA) mixed models, and methods based on frequency domain analysis, wavelet transformation or empirical mode decomposition (EMD) method. Methods based on multiple monitoring models such as in Zhou et al. [7] paper are created based on multisensory systems. In addition, the rapid development of artificial intelligence (AI) and advanced methods of inference enable more and more effective application of these methods to predict tool condition [8,9].

Hassan et al. [10] proposed a monitoring system for online prediction and prevention of tool chipping during intermittent turning. A correlation between the chipping size and cutting parameters was designed to protect the machined surfaces. The work presented an integrated system based on acoustic emission (AE) signal processing in order to detect the tool pre-failure before tool chipping, and focused on cracks due to mechanical loads during an intermittent turning operation. The TKEO-HHT (Teager Kaiser Energy Operator-Hilbert–Huang Transform) technique was used which has the ability to deal with the nonstationary and nonlinearly AE_RMS_ signal in the pre-failure phase. This method successfully predicted tool chipping before failure with a processing time of 2 ms. The determined parameter, Ψ_BW_, showed an exponential relationship with chipping, which made it possible to determine the threshold depending on the allowable chipping. The algorithm was optimized to provide sufficient time to stop the machine from damaging the workpiece. A new method of tool wear modeling is the application of a dedicated tribometer, which is able to simulate tribological conditions between the tool and workpiece. Rech et al. [11] investigated a contact pressure and sliding velocities (s_n_, V_s_) during turning. Tribological conditions were used to identify a wear model with a new tool geometry. The modeling method was based on an orthogonal cutting simulation (ALE) developed with Abaqus Implicit. In this work, the researchers reported that using the contact temperature as a parameter in the wear model was not a good idea. Instead, they decided to identify a wear model based on the contact pressure, s_n_, and sliding velocity, V_s_. Currently, this is in accordance with a trend in the field of tribology. They found that this model was very good to predict crater wear. This wear model has been implemented in numerical cutting model which is able to simulate cutting operations.

Currently, there is a lot of work that deals with monitoring wear during hard machining. One such work is Scheffer et al. [12] paper, where the researchers developed an accurate and flexible system for monitoring tool wear during hard turning. They designed an artificial intelligence (AI) model for monitoring crater and flank wear during hard turning. The purpose of developing the model was to obtain an intelligent and dynamic method. This modern approach to monitoring was based on parameters correlated directly with tool wear such as cutting force, vibration, and AE signals. In connection with this assumption, eight experiments were carried out with simultaneous measurement of cutting force, vibrations, AE, and temperature. An additional advantage of the chosen method was the ability to identify and isolate disturbances generated during the process, which was important because it was difficult to determine if the change in the sensor signal was due to wear or interference from the process. Additionally, they analyzed a self-organizing map (SOM) to identify interference that occurred during the process and they applied the method to achieve more efficient prediction of wear during hard machining. Another example of monitoring tool wear during hard turning is Ozel et al. [13], in which a neural network model was created for predicting tool wear and surface roughness. It demonstrates that the trend in process monitoring focuses not only on tool wear, but also on the evaluation of the machined surface to allow the best machining efficiency. This study utilized neural network modeling as compared with regression models. A neural network was obtained with the following seven inputs: workpiece hardness in Rockwell-C, cutting speed (m/min), feed rate (mm/rev), axial cutting length (mm), and mean values of three force components Fx, Fy, Fz (N). The small flank wear and surface roughness root mean square (RMS) errors on the test data showed the reliability of the method. The validation using neural networks gave better results than the use of regression models. The developed forecasting system was able to accurately predict surface wear and roughness. The wide range of use of aviation alloys contributed to the development of work in which tool wear is tested during machining of Inconel. One of the works is Capassoa et al. [14] investigation, in which the characteristics of tool wear during Inconel DA 718 turning with inserts with different coatings were examined. Tool wear was developed using three-dimensional (3D) volumetric wear progression. A predictive model was created based on both 3D and flank wear patterns. The model of tool wear with TiAlCrN/TiCrAl_52_Si_8_N PVD coating and AlTiN at different cutting speeds reached a value of the fitting factor R of over 93%, which meant that the method produced very good results. In addition to the new predictive models, they found that the tool with a PVD nanocomposite coating exhibited a substantial reduction in chipping, which confirmed superior wear resistance. In summary, the applied methodology proved that the volumetric wear prediction method was reliable.

There is a lack of studies comparing the use of different measured quantities as input data during the cutting process. If they already appear, it does not interfere with the network structure such as by changing the activation function or the number of neurons in the layer. In particular little information relates to the processing of hard materials, where the most advantageous information is about predicting tool wear. Therefore, studies have focused on comparing the effectiveness of predicting neural network models with different structures and with different input data.

In this paper, artificial neural networks are presented to predict tool wear based on various input data such as cutting forces and mechanical vibrations. Measurements of selected physical quantities were carried out during turning of hardened steel with constant cutting parameters.

## 2. Materials and Method

The turning of hardened bearing steel 100Cr_6_ with a hardness of 61 ± 1 HRC was conducted. The tool material was oxide ceramics (Al_2_O_3_ + TiN). Mechanically fixed inserts SNGN120408 MC2 (Kennametal, Latrobe, PA, USA) were used for testing. The research was carried out on a universal lathe TUR560E (FAT, Wroclaw, Poland) with constant cutting parameters:cutting speed *v_c_* = 180 m/min;rotational speed *n* = 1400 rev/min;feed *f* = 0.08 mm/rev;depth of cut *a_p_* = 0.1 mm.

After each pass (length of the shaft *L* = 150 mm and cutting time of a single pass *t_s_* = 1.34 min) the value of flank wear, *VB_c_*, was measured (*VB_c_*, flank wear of the tool corner), by means of a workshop microscope with a resolution of 0.01 mm.

During the turning operation, the following cutting force components were measured:*F_x_*, *F_f_* for feed direction;*F_y_*, *F_p_* for radial direction;*F_z_*, *F_c_* for main direction,

In addition, the acceleration of vibration was measured in the following different directions:*A_x_*, *A_f_* for feed direction;*A_y_*, *A_p_* for radial direction;*A_z_*, *A_c_* for main direction.

Figure 1 presents a simplified diagram of the measurement setup, which considers the location of sensors and additional components necessary for signal processing and analysis. Piezoelectric sensors were used to measure cutting forces and mechanical vibrations.

The maximum, minimum, and mean square values of cutting forces and vibration accelerations were selected as diagnostic measures. The mechanical vibrations were measured by a piezoelectric three-axis acceleration sensor fixed to the toolholder using a thread, while the cutting forces were measured using a piezoelectric measuring platform.

On the basis of digital signals sent to the computer, the mean square RMS values (Equation (1)) were evaluated:(1)$$M_{RMS}=\sqrt{\frac{1}{T_{2}-T_{1}}\overset{T_{2}}{\underset{T_{1}}{\int }}{\left[x\left(t\right)\right]}^{2}dt}$$
where *M_RMS_* is the mean square value for arbitrary diagnostic measure.

The time interval for determining the maximum, minimum, and RMS value was 4 s and the obtained measures were correlated with the corresponding tool wear values.

Under the same conditions, the wear process was carried out for 15 tool tips (15 tests). For each corner, the test was continued until the wear value *VB_c_* ≈ 0.4 mm was reached. The conventional tool life criterium that was adopted was *VB_c_* = 0.3 mm.

## 3. Results 

### 3.1. Tool Wear Analysis

Figure 2 shows the relationship between the flank wear, *VB_c_*, and the cutting time, *t_s_*, for all 15 tool corners. To determine the relation, *VB_c_* = a*t_s_*^3^ + b*t_s_*^2^ + c*t_s_*, a third degree polynomial function was selected as the most representative for the tool wear process. This function reflects the results obtained in the best way, and the coefficient R^2^ = 0.98, which indicates a high adjustment to the selected mathematical function.

The graph shows that the assumed tool life criterion, *VB_c_* = 0.3 mm, is reached for *t_s_* = 25 min. This criterion was selected based on previous experience related to the machining of hardened steels. Above this value, the probability of chipping of ceramic tool increased significantly, due to an increase in the level of mechanical vibration amplitudes.

The next step was to recognize the relationship between the tool wear and designated measures of diagnostic signals. Figure 3 depicts an example of the relation between the maximum value of the feed force, *F_f_max_,* and tool wear, *VB_c_*. This relationship was described by the linear function *F_f_max_* = a⋅*VB_c_* + b and the coefficient R^2^ = 0.78. For all other diagnostic measures, based on force measurements (i.e., *F_i_max_*, *F_i_min_*, and *F_i_RMS_*), the best results were also obtained for the linear function.

However, there are different dependencies of the type, *F_i_* = a⋅*VB_c_* + b, when the data will be divided into individual tool corners. The R^2^ coefficient indicates correct matching of the assumed mathematical function to the experimental results. Figure 4a,b shows the individual R^2^ coefficient for all analyzed tool tips and for two exemplary measures, *F_f_max_* and *F_p_max_*.

For example, for the *F_p_max_* measure, the extreme waveforms were selected, i.e., for the extreme values of R^2^, in order to illustrate changes in the amplitudes of the diagnostic measure as a function of tool wear (Figure 5). The analyzed changes of the *F_p_max_* measures are described by a linear function with similar coefficients but with different dispersion of results. The larger the spread of results (the smaller the R^2^), the more difficult it is to build a correct diagnostic model, although in the case of cutting forces the best results were obtained (the highest R^2^ coefficient).

The dependencies based on accelerations of vibrations look different and the dispersions of results are much larger. Figure 6 shows R^2^ values for all tool tips and for exemplary measures, *A_f_* and *A_p_*. In comparison to cutting forces, extreme values are much higher and equal to ΔR^2^ = 0.545 for *A_f_* and ΔR^2^ = 0.49 for *A_p_*, respectively. These differences are illustrated in Figure 7. In the worst case, for *A_p_* − R^2^ = 0.34. The matching of the mathematical function to the actual results is unsatisfactory. The same analysis was carried out for all measures (in the X, Y, and Z directions), obtaining similar relationships.

### 3.2. Diagnostic Model in the Form of a Regression Model

Section 3.1 describes the changes in diagnostic measures (for cutting forces and acceleration of vibrations) as a function of tool wear. The main purpose of the diagnosis of the cutting tool’s condition is to recognize its degree of wear precisely based on the measured values of forces and vibrations. Nevertheless, a different approach to this issue can be applied. In industrial practice, very often two states are recognized, acceptable and unacceptable, i.e., when the tool tip should be replaced with a new one. For this purpose, the permissible tool wear value must be defined, i.e., tool life criterion. This work assumes: *VB*_c_ < 0.3 mm (a tool tip capable of machining) and *VB_c_* ≥ 0.3 mm (a blunt tool tip).

Figure 8 shows the two tool conditions (for all 15 tool corners) for two measures of cutting forces, *F_p_max_* and *F_f_max_*. Separation of the two areas, a tool tip capable of machining and a blunt tool tip, is the task of the monitoring system, which works based on various mathematical algorithms.

Such an analysis can be carried out because of several diagnostic measures that were selected in this work. The task is not complicated and in the diagnostic systems it is called the classification. However, a two-step evaluation of the tool condition is not always enough. Usually, recognition of the tool condition in the next cycle is the relevant information to withdraw the tool before exceeding the allowable wear. In this situation, we are dealing with prediction, and therefore a valid mathematical model for prediction that can assess the tool condition at any time. The simplest model is the one-variable regression equation shown in Figure 9.

For cutting forces components, the linear relationship of the type *y* = a⋅*x* + b (i.e., *VB_c_* = a⋅*F_i_* + b) is best suited for assessing tool wear. In this context, it is enough to substitute the appropriate value of the cutting force component and read the value of the tool wear indicator. A similar procedure is applied when using diagnostic measures based on mechanical vibration signals. The only difference is that for vibrations the best-suited dependence is the logarithmic function of the type, *y* = a⋅ln(*x*) + b (i.e., *VB_c_* = a⋅ln (*A_i_*) + b, Figure 10). The basic disadvantage of the one-variable regression model is low precision. Figure 10b shows an example, where the coefficient of determination, R^2^ = 0.34, and dispersions of test results are very large. Therefore, it is difficult to carry out the correct verification process. The natural dispersion of experimental results means that the predicted values are not precise and are sometimes burdened with error. Hence it is better to use multivariable models or artificial intelligence algorithms, such as artificial neural networks.

### 3.3. Diagnostic Verification Based on Artificial Neural Networks

After the analysis of individual diagnostic measures, neural network models were developed to recognize tool wear using multilayer perceptron (MLP) feedforward networks. The input data were diagnostic measures based on the analysis of vibration accelerations and the components of the cutting force. The structure of the used networks, on the entry of two or four neurons, in the hidden layer the number of neurons, varied from four to 20, while in the initial layer it was one neuron. The first step was focused on learning the neural networks using the Broyden—Fletcher—Goldfarb—Shanno algorithm (BFGS), which is considered one of the most effective. Input data were results for 13 tool tips working with identical cutting parameters. However, the results from the next two cutting edges were used to validate the developed neural networks. In order to select the best network, the activation functions were changed in the hidden and the initial layer: linear, logistic, hyperbolic, and exponential. The functions used are listed in Table 1.

#### 3.3.1. Models of Neural Networks Used to Recognize the VBc Tool Wear Based on Accelerations of Vibrations

In the first place, the results of vibration accelerations were analyzed starting from the network learning stage. The networks created with different configuration of the activation functions in the hidden layer and the output layer. Among the 130 models developed, three models with the best learning and testing efficiency were selected. Table 2 shows the MLP neural networks together with a list of all parameters generated in the Statistica program. After entering data into the program, the number of random samples was assumed at the level of 70% for the training set, 15% for the test set, and 15% for the validation set.

The network number 126 has the best quality of testing around 85%, while the network number 23 has the best learning efficiency of approximately 88%. The number of neurons in the hidden layer, the number of cycles of the BFGS algorithm (Broyden–Fletcher–Goldfarb–Shanno algorithm, MLP network learning algorithm), and the function of the hidden layer and the initial layer affect the effectiveness.

The *SOS* error function, *E_SOS_* (sum of squares), was used to determine the error during learning, testing, and validation. The error function determines the correspondence between calculated values and actual values. In Statistica, the error function is calculated as the sum of squared differences based on the Equation (2):(2)$$E_{SOS}=\overset{n}{\underset{i=2}{\sum }}{\left(y_{i}-t_{i}\right)}^{2}$$
where *n* is the number of examples (input and output pairs) used for learning, *y_i_* is the network prediction (network output), and *t_i_* is the “real” value (output according to data) for *i*th value.

The effectiveness of selected neural networks was evaluated based on the root mean square error (RMSE). After learning the network, new data was introduced and the effectiveness of the prediction of the tool wear, *VB_c_,* was checked. A comparison of the effectiveness of individual networks based on new data is presented in Table 3.

The smallest error was obtained for a network with 20 neurons in the hidden layer (MLP 2-20-1). The mean square error using this network was 0.049. Figure 11 graphically illustrates the correlation between experimental data and the predicted data for the MLP 2-20-1 networks.

#### 3.3.2. Models of Neural Networks Used to Recognize the VBc Tool Wear Based on the Cutting Force Components

The next step in checking the effectiveness of tool wear prediction value was the development of an ANN model with input values of the components of cutting forces. Two diagnostic measures were selected for the creation of the model, *F_p_max_* and *F_f_max_*. The 95 different models of neural networks were created corresponding to models based on vibration accelerations. Table 4 presents selected models for analyzing the effectiveness of tool wear prediction.

The best learning quality was obtained for a network with twenty neurons in the hidden layer (No. 94), approximately 93%. Similarly, the same network achieved the best testing efficiency, approximately 95%. Noteably, by increasing the number of neurons in the hidden layer, the performance of the model was improved. It is obvious that for the networks with the best training and testing quality, the smallest RMS error was obtained. Table 5 presents the comparison of the obtained RSM errors for individual models. 

However, Figure 12 shows the dispersion of predicted and experimental values for the best performing model.

#### 3.3.3. Models of Neural Networks Used to Recognize the VBc Tool Wear Based on Four Variables

The final stage of the analysis was the creation of a model of neural networks based on four different variables. Diagnostic measures with the best correlation coefficient R^2^ were selected as input data. Two measures of the cutting forces component were selected, *F_p_max_* and *F_f_max_*, and two measures of vibration acceleration, *A_f_* and *A_p_*. Thirty different models of neural networks were created. Table 6 presents three models selected for analyzing the effectiveness of tool wear prediction.

Table 7 lists the root mean square error RMSE for three selected networks. However, Figure 13 presents the spread of results between the predicted and experimental values of tool wear based on the MLP 4-6-1 model (No.21). 

The best learning quality, approximately 97%, was obtained for the network No. 5, while the best test efficiency was a network with six neurons in the hidden layer of about 98% (No. 21). Similar to the previous analysis, after selecting the best quality models, new data were introduced to determine the effectiveness of the neural networks at the validation stage. The smallest RMS error was obtained for a network with six neurons in the hidden layer and a linear function of activation in the hidden layer of 0.040 (No.21). In this analysis, a significant impact of the activation function used on the network performance was noticed. The error values from all analyzed models are summarized in Figure 14.

On the basis of all the developed models of neural networks, the best network performance for four different variables was determined (for diagnostic measures based on vibration and cutting force signals). This solution has a practical drawback because it requires measurement of two physical quantities, vibration and forces. In practice, systems based on measurements of several physical quantities are problematic and cost intensive. Therefore, sometimes a better compromise is the use of simple measuring methods (as in the case of vibrations) in exchange for greater effectiveness in recognizing the condition of the cutting tool. There are some limits of compromise, but this should be checked in practical applications.

## 4. Conclusions

This paper presents a neural network model to predict tool wear based on cutting forces and mechanical vibrations. From this study, the following conclusions can be drawn:The correlation coefficient between tool wear, *VB_c_*, and the diagnostic measure assumes different values depending on the direction of vibration or force measurement. The best correlation coefficient was obtained for the radial cutting forces *F_p_max_*. The coefficient value R^2^ = 0.87 indicates a good correlation with the power mathematical function. On the basis of the coefficient, the input data for creating the network was also selected.Proper selection of the number of neurons in the hidden layer and activation function in the hidden and initial layers significantly affect the effectiveness of predicting the tool wear value. Changes in the structure of the model at the beginning of its creation by the user help to achieve prediction at a satisfactory level. The artificial neural network, MLP, is an effective model for predicting tool condition during machining difficult-to-cut materials. The use of various diagnostic measures increases the efficiency of prediction.The correlation coefficient obtained in the analysis of vibration accelerations was definitely lower than in the analysis of cutting forces. Nevertheless, the tool wear model, ANN, based on the measures of acceleration of vibrations *A_p_* and *A_f_* obtained the ability to forecast tool wear with the efficiency loaded by the mean square error RMSE = 0.049.Wear prediction based on measurements of cutting force components *F_p_max_* and *F_f_max_* obtained slightly better results than at vibration accelerations. The error accomplished was RMSE = 0.045 mm. This means that both cutting forces and vibration acceleration are equally good for assessing tool wear during machining difficult-to-cut materials.By creating different structures of the ANN model, the most effective prediction possibility for the model with the four input measures was obtained: *F_p_max_*, *F_f_max_*, *A_p_* and *A_f_*. The use of various diagnostic measures produced the best prediction results, RMSE error = 0.040 mm. In this case, the tangent activation function in the hidden layer and the linear activation function in the output value accomplished the best effects.

## Figures and Tables

**Figure 1 materials-12-03091-f001:**
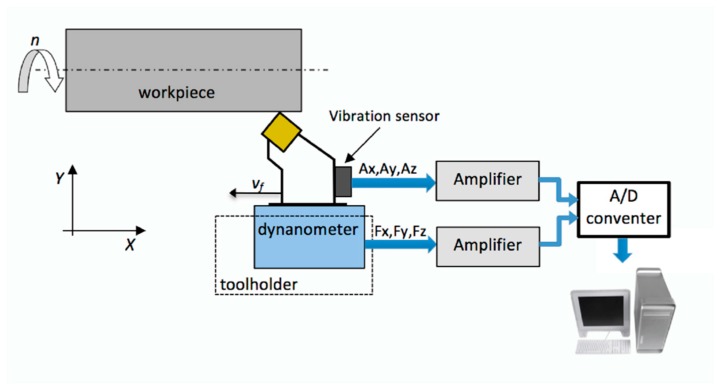
Scheme of measurement line used during the tests.

**Figure 2 materials-12-03091-f002:**
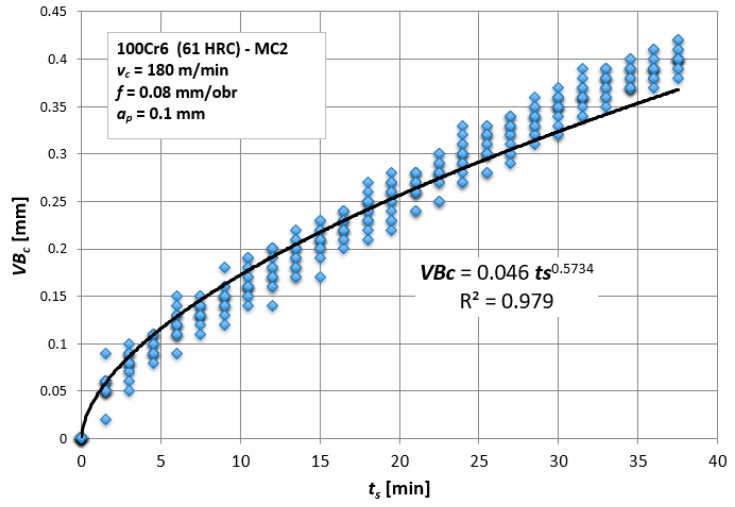
Tool wear, *VB_c_*, as a function of cutting time, *t_s_*, including all tests carried out.

**Figure 3 materials-12-03091-f003:**
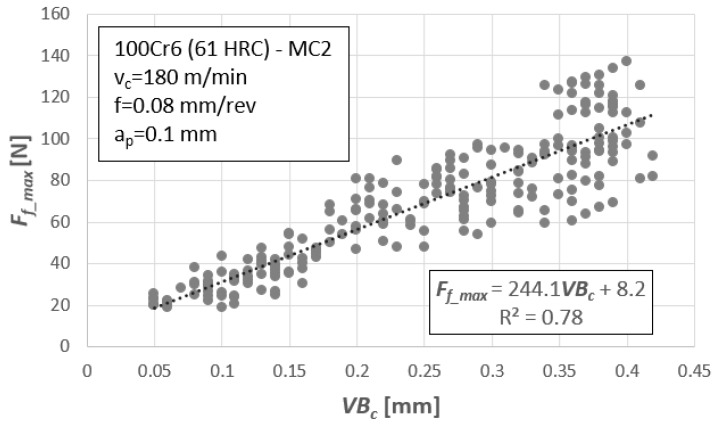
Feed force, Ff_max, as a function of flank wear, *VBc*, for all 15 cutting wedges.

**Figure 4 materials-12-03091-f004:**
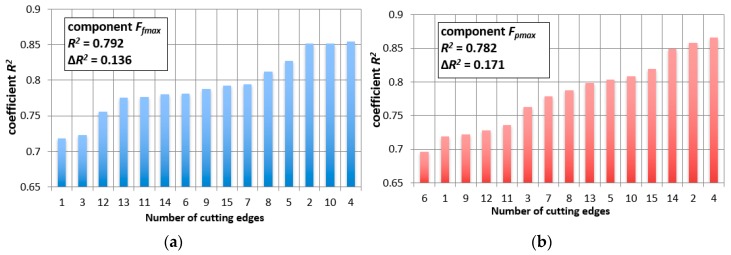
Matching coefficient, R^2^, for the: (**a**) F_f_max_ measure and (**b**) F_p_max._

**Figure 5 materials-12-03091-f005:**
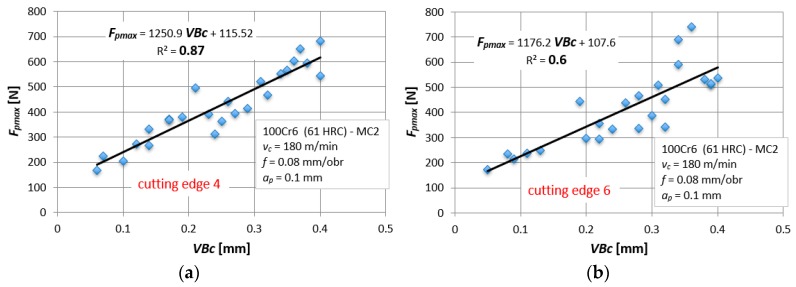
Cutting force, Fp_max, as a function of the tool wear, VBc, for tool tip (**a**) No. 4 and (**b**) No. 6.

**Figure 6 materials-12-03091-f006:**
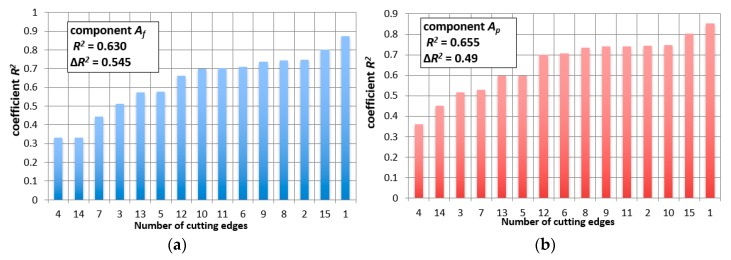
Matching coefficient R^2^ for (**a**) the Af measure and (**b**) the Ap measure

**Figure 7 materials-12-03091-f007:**
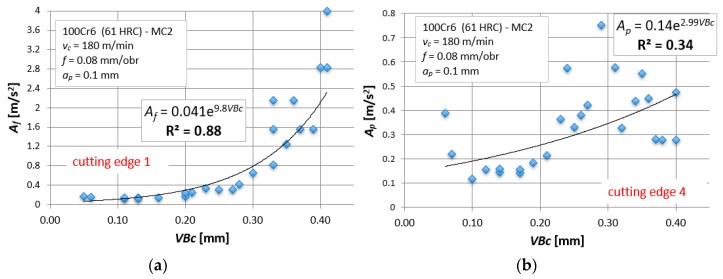
Acceleration of vibrations in feed direction, Af, as a function of the tool wear, VBc, for tool tips (**a**) No. 1 and (**b**) No. 4.

**Figure 8 materials-12-03091-f008:**
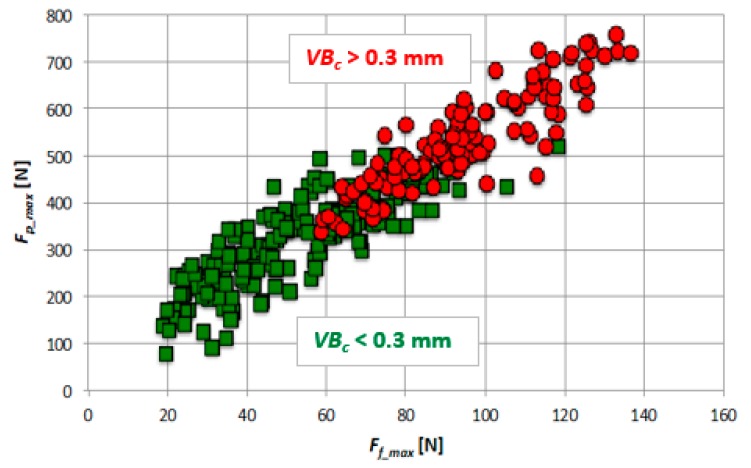
Two tool conditions for Fp_max and Ff_max measures (for 15 tool corners).

**Figure 9 materials-12-03091-f009:**
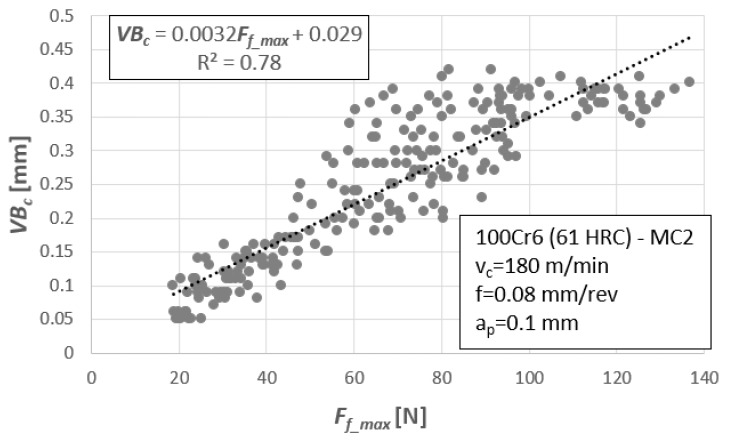
Diagnostic model in the form of a regression model for 15 tool tips.

**Figure 10 materials-12-03091-f010:**
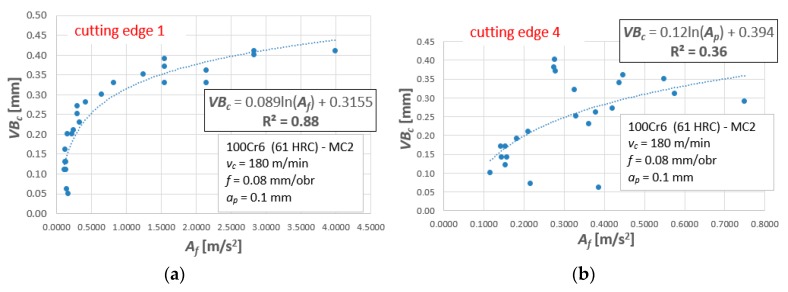
Exemplary one-variable regression model for Af: (**a**) tool corner No. 1 and (**b**) tool corner No. 4.

**Figure 11 materials-12-03091-f011:**
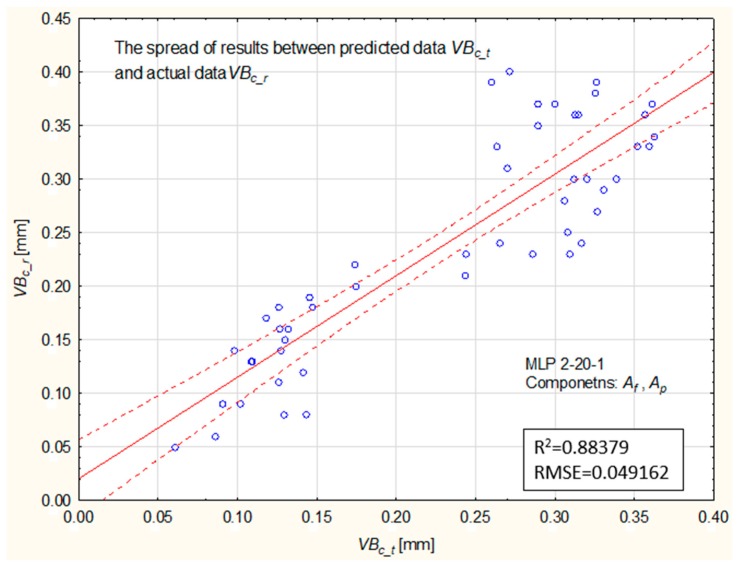
Comparison of predicted values, VBc_t, with experimental data, VBc_r, at the validation stage (MLP 2-20-1).

**Figure 12 materials-12-03091-f012:**
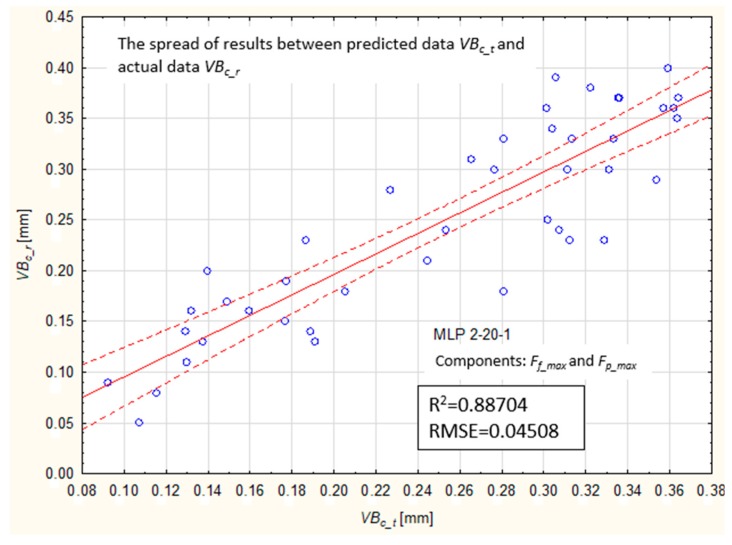
Comparison of predicted values, VBc_t, with experimental data, VBc_r, at validation stage (MLP 2-20-1).

**Figure 13 materials-12-03091-f013:**
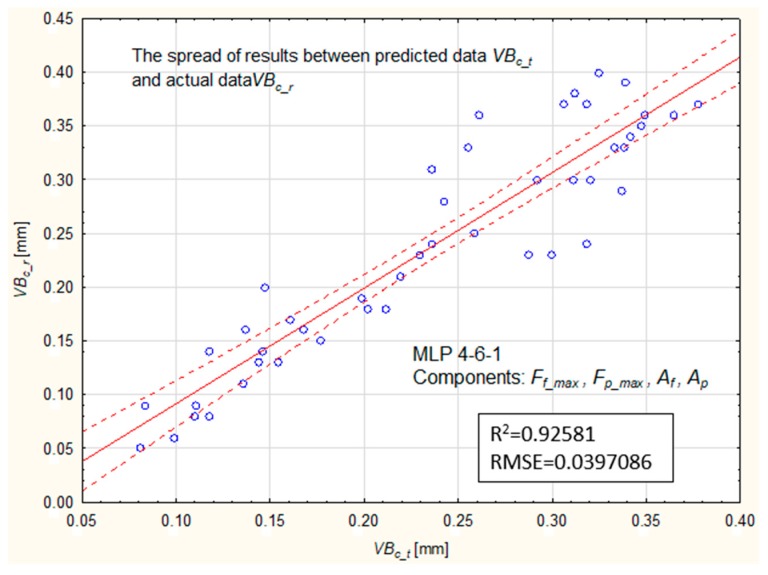
Comparison of predicted values, VBc_t, with experimental data, VBc_r, at the validation stage MLP 4-6-1 (based on Fp_max, Ff_max, Af, and Ap variables).

**Figure 14 materials-12-03091-f014:**
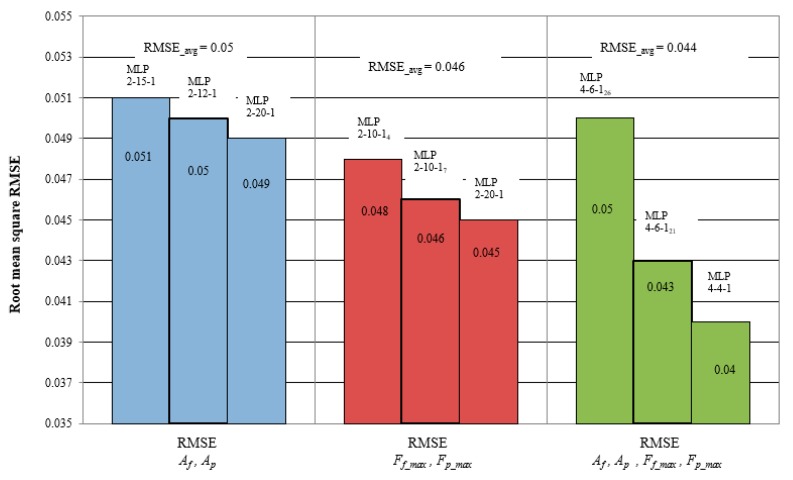
Comparison of RMS errors of the neural networks used in the validation stage.

**Table 1 materials-12-03091-t001:** Activation functions applied at the network learning stage.

Function	Equation
Linear	x
Logistics	$\frac{1}{1-e^{-1}}$
Hyperbolic (Tanh)	$\frac{e^{x}-e^{-x}}{e^{x}+e^{-x}}$
Exponential	$e^{-x}$

**Table 2 materials-12-03091-t002:** Selected neural networks based on vibration acceleration analysis (generated in Statistica).

**No.**	**Network Name**	**Quality (Learning)**	**Quality (Testing)**	**Quality (Validation)**	**Activation Function (Hidden Layer)**	**Activation Function (Output Layer)**
23	**MLP 2-12-1**	0.8826	0.8336	0.9119	Logistic	Exponential
72	**MLP 2-15-1**	0.8818	0.8449	0.9249	Exponential	Exponential
126	**MLP 2-20-1**	0.8797	0.8518	0.9188	Tanh	Exponential
**No.**	**Network name**	**Learning algorithm**	**Error (learning)**	**Error (testing)**	**Error (validation)**	**Error function *E_sos_***
23	**MLP 2-12-1**	BFGS 55	0.0012	0.0010	0.0007	*SOS*
72	**MLP 2-15-1**	BFGS 84	0.0008	0.0007	0.0007	*SOS*
126	**MLP 2-20-1**	BFGS 68	0.0007	0.0007	0.0007	*SOS*

**Table 3 materials-12-03091-t003:** Root mean square error (RMSE) values for three MLP neural network (based on acceleration of vibrations).

Network Name	RMSE
MLP 2-12-1	0.051
MLP 2-15-1	0.050
**MLP 2-20-1**	**0.049**

**Table 4 materials-12-03091-t004:** Selected neural networks based on the cutting force components (generated in Statistica).

No.	Network Name	Quality (Learning)	Quality (Testing)	Learning Algorithm	Activation Function (Hidden Layer)	Activation Function (Output Layer)
4	**MLP 2-10-1**	0.8787	0.9245	BFGS 4	Linear	Linear
7	**MLP 2-10-1**	0.9228	0.9502	BFGS 149	Tanh	Linear
94	**MLP 2-20-1**	0.9259	0.9510	BFGS 140	Tanh	Logistic

**Table 5 materials-12-03091-t005:** The RMSE values for three MLP neural network (based on the cutting force components).

Network Name	RMSE
MLP 2-10-1 (4)	0.048
MLP 2-10-1 (7)	0.046
**MLP 2-20-1**	**0.045**

**Table 6 materials-12-03091-t006:** Selected neural networks based on four different variables (generated in Statistica).

No.	Network Name	Quality (Learning)	Quality (Testing)	Learning Algorithm	Activation Function (Hidden Layer)	Activation Function (Output Layer)
5	**MLP 4-4-1**	0.9711	0.9766	BFGS 90	Tanh	Exponential
21	MLP 4-6-1	0.9697	0.9817	BFGS 60	Tanh	Linear
26	**MLP 4-6-1**	0.9706	0.9807	BFGS 62	Tanh	Tanh

**Table 7 materials-12-03091-t007:** The RMSE values for three MLP neural network (based on four variables).

Network Name	RMSE
MLP 4-4-1	0.043
**MLP 4-6-1 (21)**	**0.040**
MLP 4-6-1 (26)	0.050

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
