# Peer review of "Prediction of Tool Wear Using Artificial Neural Networks during Turning of Hardened Steel"

_materials, 2019, doi:10.3390/ma12193091_

Round 1

Reviewer 1 Report

The authors show careful consideration on the assessment of many test data. Hence it is recommended to publish to Materials.

Author Response

Thank you so much for your review report. I'm sending you corrected manuscript.

Best regards, 

Martyna Wiciak-Pikuła 

Reviewer 2 Report

Nothing to add!

Author Response

Thank you so much for your review report. I'm sending corrected manuscript.

Best regards, 

Martyna Wiciak-Pikuła

Reviewer 3 Report

The manuscript entilted “Prediction of tool wear using Artificial Neural Network during turning of hardened steel” was presented by  Paweł Twardowski and Martyna Wiciak-Pikuła. Some mistakes should be avoided in the manuscript, and I do suggest the authors to read the MS thoroughly and make the necessary changes. Some examples are given as below:

1.     Introduction section needs to be rewritten. The current literature review is not sufficient. Please mention more studies related to the matter of the article in the section of Introduction. Please give the research gap and the aims of the current study.

2.     “feed f = 0,08 mm/rev” should be “feed f = 0.08 mm/rev”. Please revise this kind of issue in the whole manuscript, such as Figs. 2-19 and Tables 2-7.

3.     Please separate “Results and discussion” to section “Results” and section “Discussion”.

4.     Please combine some figures to one figure and use (a), (b) for the sub-figures, such as Figs. 4,5; Figs. 6,7; Figs. 8,9; Figs. 10,11; Figs. 14,15; Figs. 17,18.

5.     Conclusion needs to be rewritten. Please provide the significance of this study.

Author Response

Thank you so much for your review report. I am sending you corrected manuscript.

1.The introduction  has been improved and supplemented with an additional literature review.

2.All numbers with "commas" have been converted into "dots".

3.The section results has been separated.

4.The drawings have been corrected in accordance with the recommendations.

5.The conclusion has been improved.

Best regards,

Martyna Wiciak-Pikuła

Reviewer 4 Report

This is a report of an exercise that looks at the various signals that are gathered during wear of cutting tools, and correlates these signals through neural network processing to the wear that occurs.

As such there is no development of understanding of the actual mechanisms of wear that occur, and therefore no understanding of how this wear could be reduced by amending the ciomposition or make up of the tool material.

As such the paper potentially provides a way to give a diagnostic technique that can be used to monitor wear of tools, but it does not address the larger goal of better development of materials understanding.

Could the authors please supply information to address this lack, and in any case comment on these points.

Author Response

Thank you so much for review report. I'm sending you corrected manuscript.

The article has been enriched with an additional literature review and mention of the deficiencies in modern literature regarding the diagnosis of hard materials. With this material, it is not necessary to reduce the wear as a value only to replace it in a timely manner before breaking the cutting tool. Based on the signals measured during the process and then the creation of the ANN model, it is possible to predict at which signal increase the tool is not suitable for further work. I hope that I have been able to answer your doubts.

Best regards, 

Martyna Wiciak-Pikuła

Reviewer 5 Report

Review of materials-535861-peer-review-v1: “Prediction of tool wear using Artificial Neural Network during turning of hardened steel”

The subject of the paper is relevant with the topics of the journal, and could offer a great deal of industrial value. Nevertheless, a number of issues reduce the value of the submission:

·         The number of references used should be increased for a paper in this area.

·         The number of experiments for such a research work is limited and should be increased.

·         Probably this is the reason that the R2 coefficient in a number of cases is below 0.8 which is expected to offer relative reliable results.

·         Table 2 should contain all the models calculated with details and comments in the text per family of models based on selected criteria.

·         Table 4 should contain all the models calculated with details and comments in the text per family of models based on selected criteria

My proposal to the editor is to reject the paper.

Author Response

Thank you so much for review report. I'm sending you the corrected manuscript.

1.Literature review has been supplemented with additional knowledge of current articles.

2.The work involved 15 experiments for the same cutting conditions. I think that at this moment it is enough to enter data into the ANN network. In the future, you can actually think about doing more tests on subsequent tests.

3.Table 2 and 4 show the models that give the best performance for this case. Other models did not give any results, and the effectiveness of the prediction had no effect, therefore the tables did not include additional data that would not be able to provide an effective prediction of the blade state.

I hope that my comments and the improved article are satisfactory.

Best regards, 

Martyna Wiciak-Pikuła

Round 2

Reviewer 4 Report

This is an interesting and useful paper

Author Response

Thank you for your review. I am grateful for the comments sent.
Best regards,
Martyna Wiciak-Pikuła

Reviewer 5 Report

In my opinion, I do not think there is a considerable improvement on the soundness of the paper when reviewing the revised version.

I must insist in my initial decision.

Author Response

Thank you for your review.
Additional corrections have been made in the introduction,
I hope the article is now more correct.

Best regards,
Martyna Wiciak-Pikuła